# Protective Role of Galanin during Chemically Induced Inflammation in Zebrafish Larvae

**DOI:** 10.3390/biology10020099

**Published:** 2021-01-30

**Authors:** Natalia Nowik, Anna Przyborowska, Waldemar Sienkiewicz, Piotr Podlasz

**Affiliations:** 1Department of Animal Sciences and Health, Institute of Biology (IBL), Leiden University, 2333BE Leiden, The Netherlands; natalia.nowik@outlook.com; 2Department of Animal Anatomy, Faculty of Veterinary Medicine, University of Warmia and Mazury, Oczapowskiego 13, 10-719 Olsztyn, Poland; sienio@uwm.edu.pl; 3Department of Pathophysiology, Forensic Veterinary and Administration, Faculty of Veterinary Medicine, University of Warmia and Mazury, Oczapowskiego 13, 10-719 Olsztyn, Poland; anna.jakimiuk@uwm.edu.pl

**Keywords:** zebrafish, galanin, inflammation

## Abstract

**Simple Summary:**

The neuropeptide galanin is known for its protective role in the central nervous system, but not much is reported about its involvement in innate immunity. Our studies showed that galanin expression became upregulated after incubation in copper sulfate, which caused neuromast damage on the lateral line of the fish. Galanin presence protected the cells from the degenerative effects of the copper sulfate incubation compared to the galanin knockout larvae. Moreover, galanin treatment resulted in a higher expression of some inflammatory markers such as *irg1l*, *il1b* and *tnfa*, which indicated its possible role in immune responses and inflammatory processes. On the other side, galanin knockout caused more severe damage to the neuromasts, as well as downregulation in expression of the studied genes. Taken together, our results showed that galanin’s function was not only limited to the central nervous system as it was involved in other processes in the organism.

**Abstract:**

During a pathological condition, many different systems are involved in the response of an affected organism. Galanin is considered to be a neuropeptide that plays an important role in the central nervous system; however, it is involved in many other biological processes, including the immune response. During our studies, we showed that galanin became upregulated in zebrafish larvae when exposed to copper sulfate. Moreover, the presence of normal levels of galanin, administration of a galanin analog NAX 5055 or galanin overexpression led to lowered lateral line damage and enhanced expression of inflammatory markers compared to the knockout larvae. The results showed that the neuroendocrine system acts multifunctionally and should be considered as a part of the complex neuro–immune–endocrine axis.

## 1. Introduction

Although a connection between the neuroendocrine and immune system was a doubtful matter for many years, it has been recently proven that they are complementary to each other [1,2]. This link is regulated by a number of neuropeptides, which play different roles in various biological processes, including innate immunity [3,4,5,6].

Galanin is a 29/30 amino acid long neuropeptide that is a major signaling peptide and belongs to the galanin family that forms the galaninergic system. The galaninergic system is a specific signaling system that plays a role in neurotransmission and neuromodulation that was first described by Tatemoto et al. [7]. The galanin peptide is widely distributed in the central and peripheral nervous tissues in various species [8], where it is involved in many biological processes such as sleep, nociception, learning and memory, depression, feeding, pituitary hormone release, stress and anxiety, osmotic regulation and water intake, thermoregulation, reproduction, metabolism and inflammation [9]. Galanin likely plays an important role in immunity, as its expression increases during inflammation [10]. Studies conducted by Kofler et al. [11] showed that galanin is involved in skin immunity, as its expression was found to be upregulated in the inflamed tissue. Another study underlined the use of galanin peptides as an antimicrobial agent against *Candida albicans* [12]. Galanin plays an important part in poststroke inflammation, including the secretion of inflammatory cytokines and leukocyte/monocyte infiltration in the brain [13]. Previous studies showed that an increase in GalR1, one of the galanin’s receptors, was closely related to upregulation of nuclear factor NF-κB, which is an important inflammatory factor [14].

Larval zebrafish has become a widely used model to study various systems, mainly because of its transparency and visualization of fluorescent processes in vivo. It has been established as an excellent model organism for studies of vertebrate biology, genetic, embryonal development, infections and central nervous system disorders, including Parkinson’s disease, schizophrenia, Alzheimer’s disease and depression [15]. Teleost fish species, such as the zebrafish, have a well-developed immune system, both innate and adaptive, which is similar to the mammalian system [16]. Additionally, up to 4–6 weeks post fertilization (wpf), larvae survive with active innate immunity only [17], which gives an opportunity to study the function of the innate inflammatory response, excluding the influence of adaptive immunity. However, genes are duplicated due to fish genome duplication, which is responsible for the major differences between mammalian and zebrafish immune responses [18]. In zebrafish, galanin is connected mainly to neurodegenerative diseases [15]; however, it is also observed in pancreatic function [19] and regulation of nutrition [20]. Neuromasts are clusters of mechanosensory hair cells that are distributed on the surface along the lateral line of zebrafish. Incubation in copper sulfate (CuSO_4_) selectively damages the hair cells by inducing oxidative stress followed by cell death [21]. This toxic effect of copper sulfate on neuromasts induces extremely rapid innate immune response to copper-induced neuromast damage, followed immediately by migration of neutrophils and macrophages [21]. Furthermore, as a result of oxidative stress, reactive oxygen/nitrogen species induce inflammation through upregulation of proinflammatory gene expression that augments further development of inflammation [22].

In this short article, we would like to show the role of galanin during chemically induced inflammation in zebrafish larvae. For this purpose, we used galanin analog NAX 5055 [22], which is the most active galanin analog. It contains the Lys-Lys-Lys(palmitoyl)-Lys-NH(2) motif and shows high affinity for galanin receptors GalR1 and GalR2. Because of its features such as availability, stability and selectivity, NAX 5055 has been used in both murine [23] and zebrafish studies [19]. A zebrafish line Tg(*hsp70l:galn*) (galn-HS) with galanin overexpression was used next to the NAX 5055 treatment as an endogenous galanin source. It gave us an opportunity to compare the influence of galanin on inflammation in the zebrafish larvae and its role in the immune–neuronal axis.

## 2. Materials and Methods

### 2.1. Fish Maintenance

In the study, the wild-type *galn*+/+ (WT), *galn*−/− and Tg(*hsp70l:galn*) lines were maintained in a glass flow system with water pH 7.0 (±0.2) and a conductivity of 700–900 μS at 28 °C (±0.2 °C). The fish were kept with a 14 h light/10 h dark photoperiod and fed three times daily ad libitum with dry food and *Artemia* sp. *nauplii* [24]. Embryos were grown in E3 medium (5 mM NaCl, 0.17 mM KCl, 0.33 mM CaCl_2_, 0.33 mM MgSO_4_) containing methylene blue (Sigma-Aldrich, Saint Louis, MO, USA) and kept in an incubator at 28.5 °C in petri dishes with a maximum capacity of 50 embryos, each with a 14 h light/10 h dark period without feeding until 3 days post fertilization (dpf). Each of the *galn*+/+, *galn*−/− and Tg(*hsp70l:galn*) lines were kept in a single separated petri dish, and *galn*+/+ was treated with NAX 5055. The Tg(*hsp70l:galn*) line was kept at 37 °C each day for one hour to induce heat shock that led to overexpression of galanin levels, which was described by Podlasz et al. (2018) [25]; the same routine was applied to the other studied groups. During imaging, fish were kept under anesthesia in egg water containing 0.02% buffered 3-aminobenzoic acid ethyl ester for approximately 30 min (Sigma-Aldrich, Saint Louis, MO, USA).

CRISPR-Cas9 technology was used to generate the galanin mutant (*galn*−/−) as described [26]. The mutant has a 10-base-pair deletion in the third exon of the *galn* gene, which results in a loss-of-function allele due to a frameshift and premature stop codon. A detailed description of this mutant is in preparation for a separate publication (unpublished data).

All fish lines were housed in the fish facility of the Laboratory of Genomics and Transcriptomics, University of Warmia and Mazury in Olsztyn, Olsztyn, Poland, which was built according to the local animal welfare standards. Studies performed on early life-stage zebrafish larvae and euthanasia do not require Ethics Committee permissions, according to the European Directive 2010/63/EU and Polish law regulations O.J. of 2015, item 266. The facility has approval to keep and breed genetically modified zebrafish lines issued by the Polish Ministry of Environment, decision number 127/2017.

### 2.2. NAX 5055 Treatment

Approximately 50 zebrafish wild-type *galn*+/+ embryos were injected and treated with NAX 5055, a galanin peptide analog and nonselective galanin receptor agonist [23] that has been successfully used in zebrafish research [19]. The embryos were injected twice. The first injection with 1 nL of NAX 5055 in a concentration of 5 μg/g was performed at the first hour postfertilization into the yolk of the embryos. The second injection was performed into the blood circulation after approximately 28 hours post fertilization (hpf). The control embryos were injected twice in the same way and time as the studied group with 1 nL of 1 × PBS. The larvae were kept in E3 medium at 28 °C between the injections. During the incubation period, the larvae were kept in copper sulfate solution containing 20 μM of NAX 5055.

### 2.3. Chemically Induced Inflammation Assay

Approximately 50 zebrafish larvae at the stage of 3 dpf from each of the four groups *galn*+/+, *galn*−/−, Tg(*hsp70l:galn*) and *galn*+/+ treated with NAX 5055 were exposed to 10 μM CuSO_4_ concentration according to d’Alençon et al. [27]. The control group *galn*+/+ was kept in PBS during the same incubation period at the same temperature. After 40 min of incubation, 18–20 larvae per group were collected for RNA isolation, snap-frozen in liquid nitrogen and subsequently stored at −80 °C; a similar number of larvae was collected for orange acridine staining [27].

### 2.4. Acridine Orange Staining

Around 20 live zebrafish larvae from the same four groups *galn*+/+, *galn*−/−, Tg(*hsp70l:galn*) and *galn*+/+, treated with NAX 5055 after CuSO_4_ incubation and *galn*+/+ larvae incubated in PBS, were stained with 5 mg/mL acridine orange (acridinium chloride hemi-[zinc chloride], Sigma) in embryo medium for 10 min and washed three times for 5 min each before being anesthetized with 0.02% tricaine and mounted in agarose for confocal imaging [28].

### 2.5. Imaging

The visualization was accomplished using an LSM 700 confocal laser scanning microscope (Zeiss, Jena, Germany). Embryos were kept under anesthesia (0.02% tricaine, Sigma) in egg water during imaging. To quantify fluorescence from acridine orange in individual embryos, the fluorescent images of embryos were generated with custom-made, dedicated pixel quantification software. The procedure was performed as described in Stoop et al. [29].

### 2.6. Quantification of Fluorescent Cells

In order to quantify fluorescent cells, images of the zebrafish larvae were analyzed with specially designed, dedicated software (A.E.N., unpublished). Analysis was performed in the caudal region of the larvae to eliminate autofluorescence from the yolk. For the analysis, three types of images were taken. First, fluorescent acridine orange stained images of untreated *galn*+/+ embryos kept in PBS were used. From these images, the average background intensity in each experimental run could be established. Second, as reference images, fluorescent images of wild-type *galn*+/+ larvae incubated with CuSO_4_ were analyzed. Each reference image was thresholded by the background value obtained from the images of untreated embryos. For each image, the sum of the pixels from the fluorescent green channel above background intensity was divided by the number of embryos. This was completed for all images within a group and provided a reference value for the amount of green fluorescent pixels per embryo at the wild-type level. For the groups *galn*−/−, Tg(*hsp70l:galn*) and *galn*+/+ treated with NAX 5055 after CuSO_4_ incubation, the same calculation as the reference images was realized, and subsequently, the number of fluorescent green pixels was provided as a percentage of the wild-type level [29].

### 2.7. RNA Isolation and Quantitative PCR

Real-time PCR was performed using SYBR Green in accordance with the manufacturer’s protocol (SYBR Select Master Mix, Applied Biosystems, Foster City, CA, USA) on 7500 Fast Real-Time PCR System instruments (Applied Biosystems). The reactions for *irg1l*, *il1b* and *tnfa* were performed under the following conditions: 2 min at 50 °C, 8 min at 95 °C, followed by 40 cycles of 15 s denaturation at 95 °C and 30 s at the corresponding melting temperatures and a final melting curve of 81 cycles from 55 to 95 °C (0.5 °C increments for every 10 s). Fluorescent signals were detected at the end of each cycle. Cycle threshold values (Ct values, the cycle numbers at which a threshold value of the fluorescence intensity was reached) were established for each sample. For each sample, the Ct value was deducted from the Ct value of a control sample and the fold change of gene expression was calculated and adjusted to the expression levels of a reference gene [*peptidylprolyl isomerase Ab (cyclophilin A) (ppial)*]. Data shown are mean ± SEM of three independent experiments. Primer sequences are available in Table 1.

### 2.8. Statistical Analysis

Statistical analysis was performed using GraphPad Prism 8 (GraphPad Software, La Jolla, CA, USA) by two-tailed *t*-test (Figure 1) or two-way ANOVA followed by Tukey’s post hoc test for multiple group comparisons (Figure 2 and Figure 3). The number of larvae used in the studies was determined by power analysis (G*Power 3.1.9.7). The normality distribution of data was evaluated by the Shapiro–Wilk test. The bars represent the mean ± SEM. Statistical significance was accepted at *p* < 0.05 (95% confidence intervals); different significance levels are indicated as follows: * *p* < 0.05, ** *p* < 0.01, *** *p* < 0.001 and **** *p* < 0.0001.

## 3. Results

### 3.1. Galanin Expression Is Induced during Inflammation

We wanted to analyze how galanin expression had changed during chemically induced inflammation with copper sulfate in the zebrafish larvae. Approximately 20 zebrafish wild-type arvae at 3 dpf were incubated in 10 mM copper sulfate solution for approximately 40 min at 28 °C. After the treatment, the larvae were used for RNA isolation and quantitative PCR reaction. Our results showed that galanin expression was approximately 4 times significantly higher in the copper sulfate treated group than in the control-vehicle-treated group (Figure 1), which indicates that galanin expression increases during the inflammatory process.

### 3.2. The Effect of Chemically Induced Inflammation and Lateral Line Damage Depends on Galanin Expression

During the study, we also wanted to test how galanin’s presence would influence the level of neuromasts’ degeneration after the incubation in copper sulfate. Four zebrafish groups, *galn*+/+*, galn*−/−, Tg(*hsp70l:galn*) and *galn*+/+ treated with NAX 5055, were incubated in copper sulfate. After 40 min, the larvae were washed three times, fixed in 4% paraformaldehyde, stained with acridine orange and prepared for imaging (Appendix A). The results were confirmed by fluorescent cell quantification using pixel count software (Figure 2). We saw a significant increase in the number of fluorescent neuromasts that reached a value of 13,600 in the *galn*−/− group, which was greater than the control *galn*+/+ group, where the fluorescence was at the level of 7800 (Figure 2). On the other side, a decrease in the number of fluorescent pixels was noticed in the groups treated with endogenous or exogenous galanin compared with the *galn*−/− larvae. In the NAX 5055-treated group, fluorescence was at the level of 5400, whereas in the Tg(*hsp70l:galn*) line it was at 7300 (Figure 2).

### 3.3. Galanin Modifies the Expression of irg1l, il1b and tnfa

Approximately 18–20 embryos per group were used to analyze expression of three genes: *irg1l, il1b* and *tnfa*. We also analyzed the basal levels of these genes in the four research groups, i.e., without CuSO_4_ treatment. A pattern of similar expression was visible without significant differences between the groups (Appendix A). We used four groups to perform qPCR: copper sulfate treated *galn*+/+*, galn*−/−, Tg(*hsp70l:galn*) and *galn*+/+ incubated in NAX 5055. The vehicle-treated *galn*+/+ group was used as a control. After 40 min of copper sulfate exposure, the groups were used for qPCR reaction. *irg1l* was significantly upregulated in the NAX 5055-treated group, as well as in the Tg(*hsp70l:galn*) group, compared to the treated *galn*+/+ group. The *irg1l* expression was approximately 1.8 times higher in the NAX 5055 group, whereas in the Tg(*hsp70l:galn*) group, the expression was approximately 1.3 times higher. A significant difference was also noticed between NAX 5055 and Tg(*hsp70l:galn*) groups (Figure 3A). The downregulation was detected in the *galn*−/− group, where the expression was approximately 1.7 times lower compared to the wild-type *galn*+/+ group (Figure 3A). A similar response was noticed in the expression levels of *il1b*, which was upregulated in both galanin-treated NAX 5055 and Tg(*hsp70l:galn*) groups compared to the wild-type *galn*+/+ larvae (Figure 3B). Galanin expression in the NAX 5055-treated group was 2 times higher compared to the control and 1.4 times higher in the galanin overexpressed Tg(*hsp70l:galn*) line. No significant change was observed, however, between *galn*+/+ and *galn*−/−, as well as between the NAX 5055 and Tg(*hsp70l:galn*) groups (Figure 3B). For the third gene, we measured the expression levels of *tnfa*, where the expression pattern was similar to the other genes (Figure 3C). In the NAX 5055-treated group, the expression of *tnfa* increased approximately 2.1 times, whereas in the Tg(*hsp70l:galn*) group the expression was almost 1.3 times higher than in the wild-type *galn*+/+ group. A significant difference was also detected between NAX 5055 and Tg(*hsp70l:galn*) groups (Figure 3C). The knockout *galn*−/− group showed a significant downregulation of *tnfa*, which was approximately 1.9 times lower compared to the wild-type *galn*+/+ larvae (Figure 3C). The results showed that overexpression or treatment with galanin led to an increase in the expression of three inflammatory markers *irg1l*, *il1b* and *tnfa*, with a similar pattern between the treated groups, whereas downregulation was observed in the galanin *galn*−/− knockout mutant larvae.

## 4. Discussion and Conclusions

Galanin is considered to be mainly involved in processes that take place in the central nervous system; however, its role during an immune response should not be underestimated. Galanin signaling works via reduction in cAMP concentration and inactivation of protein kinase A (PKA) or activation of protein kinase C (PKC), which are also regulators of immune cell functions [30,31]. It has been shown that both neuronal and non-neuronal galanin or a galanin-like peptide are involved in responses [32] to various inflammatory conditions, where they modulate the expression of proinflammatory cytokines and function of immune cells [33]. During our study, we found that galanin expression was upregulated during an inflammatory process caused by a chemically induced inflammation using copper sulfate in zebrafish larvae. Our results supported previous research showing that galanin expression increased during a local inflammation. A protective role of galanin has been studied mainly with reference to central nervous system conditions. A mice line with galanin overexpression was shown to be resistant to the experimental autoimmune encephalomyelitis [34] compared to the control group that developed the condition. In our research, we used both the galanin analog NAX 5055 and a zebrafish line Tg(*hsp70l:galn*) with galanin overexpression to study the effect of galanin treatment and overproduction on the inflammatory process. We found that after CuSO_4_ incubation the number of degenerating neuromasts was significantly lower after NAX 5055 treatment and in the Tg(*hsp70l:galn*) line compared to the control group. On the other side, galanin knockout can also affect the progression of an inflammatory process. In a murine model, mutation in one of the endogenous receptors of galanin (GAL3) resulted in more severe arthritis [35], and it was found to alter psoriasis progression in mice [36]. In the galanin *galn−/−* knockout zebrafish mutant larvae, more lateral line cells were damaged by the copper sulfate effect than in the control group. These findings show that galanin activity is not only limited to the nervous system, but has a ubiquitous effect on an organism when an inflammatory process occurs. Based on these studies, we strongly believe that the protective effect of galanin overexpression is not only relevant for CuSO_4_ effect but can also be applicable to other forms of chemical wounding, chemically induced inflammation and cell degeneration. Although this mechanism of action is not yet clear, it has been shown that galanin modulates cytokine and chemokine expression in human macrophages [33]. Koller et al. [33] showed that leukocytes can serve both as a source and acceptor of galanin receptor ligands and that macrophages secrete full-length galanin and express galanin receptors. Moreover, stimulation with exogenous galanin resulted in increased CCL3 and CXCL8 levels in the treated macrophages [33]. During our research, we wanted to investigate how the incubation in copper sulfate would affect the expression of some inflammatory markers. Treatment with NAX 5055, as well as galanin overexpression in the Tg(*hsp70l:galn*) line, led to a significant increase in *irg1l*, *il1b* and *tnfa* expression. Locker et al. [36] analyzed expression of some cytokines in the GAL3 knockout mice with IMQ-induced psoriasis. The results showed the GAL3 mutant mice expressed less IL-17A (65% reduction), IL-22 (80% reduction), IL-23 (64% reduction) and TNF-α (49% reduction) compared with WT animals. Other studies showed that galanin was found to increase the expression of IL-1β up to 1.5-fold; TNF-α, IL-10, IL-18 and CCL3 up to 2-fold; and CXCL8 up to 4-fold in nonactivated monocytes [37]. On the other hand, we showed downregulation of *irg1l* and *tnfa* in the knockout *galn−/−* larvae, which indicates that galanin deficiency leads to decreased expression of cytokines and plays a role in an inflammatory process; however, its function and mechanism are not entirely clear yet.

In conclusion, galanin plays a role as a modulator in immunoregulatory mechanisms and is involved in cytokine expression. Several studies show that galanin has both anti-inflammatory and proinflammatory effects, which can be connected to complicated signaling pathways induced by the interactions between GAL and GALR. Nevertheless, galanin might be a promising therapeutic resource for several conditions, such as chemical wounding or induced inflammation; however, its possibilities require further and detailed studies.

## Figures and Tables

**Figure 1 biology-10-00099-f001:**
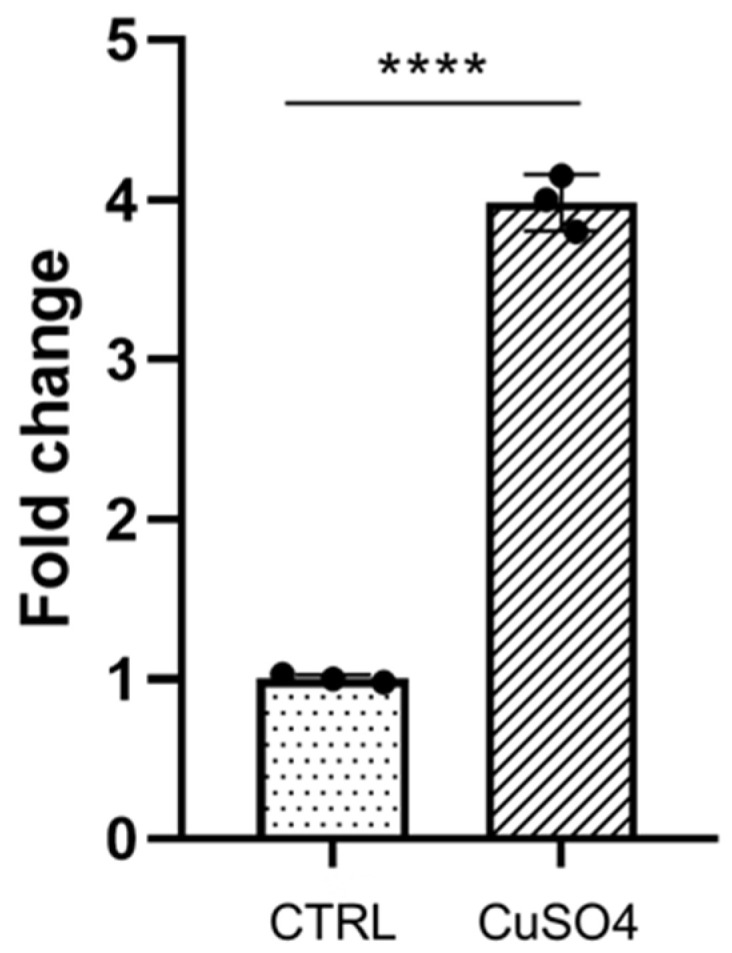
Galanin expression before (CTRL) and after copper sulfate treatment (CuSO_4_). Galanin is approximately 4 times upregulated after copper sulfate exposure compared to the untreated group. Data and mean ± SEM were pulled from three independent experiments. **** *p* < 0.0001 (determined using unpaired *t* test).

**Figure 2 biology-10-00099-f002:**
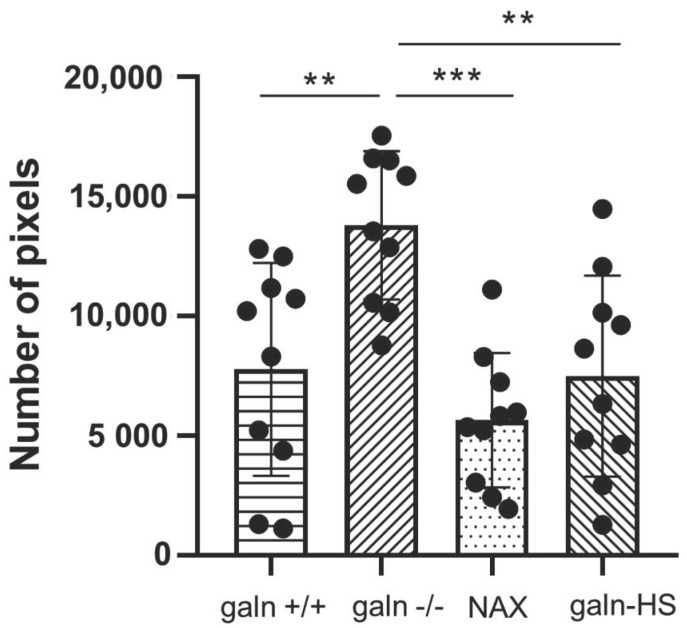
Cell degeneration after copper sulfate treatment. Number of fluorescent pixels after copper sulfate exposure in the wild-type *galn*+/+ group (*galn+/+*), a line with galanin knockout *galn*−/− (*galn*−/−) and wild-type *galn*+/+ treated with galanin analog NAX 5055 (NAX) and a line with galanin overexpression Tg(*hsp70l:galn*) (galn-HS). Data and mean ± SEM were pulled from three independent experiments. ** *p* < 0.01; *** *p* < 0.001 (determined using ANOVA with Tukey’s post hoc test).

**Figure 3 biology-10-00099-f003:**
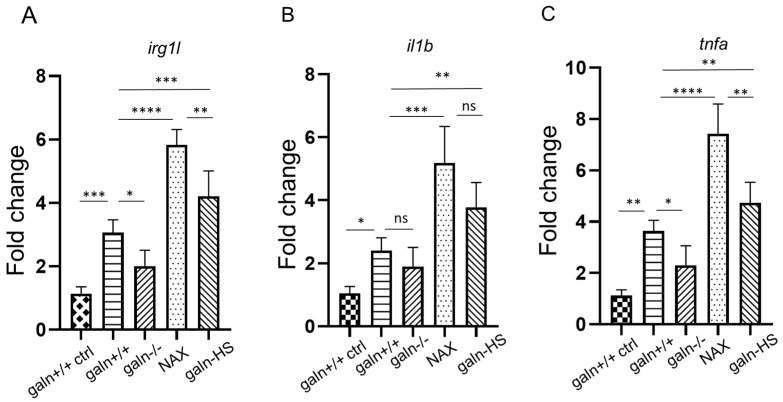
Quantitative PCR of *irg1l, il1b* and *tnfa* after copper sulfate incubation. (**A**–**C**) qPCR of *irg1l, il1b* and *tnfa* after incubation with copper sulfate in *galn*+/+*, galn−/−* and wild-type *galn*+/+ treated with galanin analog NAX 5055 (NAX) and a line with galanin overexpression Tg(*hsp70l:galn*) (galn-HS) groups was performed at 3 dpf. Control wild-type *galn*+/+ larvae (*galn*+/+ ctrl) were incubated only in PBS without copper sulfate. Data and mean ± SEM were pulled from three independent experiments. * *p* < 0.05; ** *p* < 0.01; *** *p* < 0.001; **** *p* < 0.0001 (determined using ANOVA with Tukey’s post hoc test); ns, not significant.

**Table 1 biology-10-00099-t001:** Primer sequences and GenBank accessions for genes analyzed in this study.

Gene	Forward 5′-3′	Reverse 5′-3′	Accession
*irg1l*	GGTTAGAAGCAAGTCCTC	TGTGTTCATCCTCCTCAG	NM_001077607
*il1b*	GAACAGAATGAAGCACATCAAACC	ACGGCACTGAATCCACCAC	NM_212844
*tnfa*	AGACCTTAGACTGGAGAGATGAC	CAAAGACACCTGGCTGTAGAC	NM_212829
*galn*	AAGGATACTCCCAGTGCAAGG	CTTTCCTGCCAGTCCGTGTT	NM_001346239
*ppial*	ACACTGAAACACGGAGGCAAAG	CATCCACAACCTTCCCGAACAC	AY391451

## Data Availability

The raw data presented in this study are available as non-published materials.

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
