# Peer review of "Protective Role of Galanin during Chemically Induced Inflammation in Zebrafish Larvae"

_biology, 2021, doi:10.3390/biology10020099_

Round 1
Reviewer 1 Report
Dear authors,
Your study highlights the role of galanin in the immune response when facing a wound inflammation. Although the role of galanin in the immune system is not new, as you acknowledge, this study is one more push to study this perspective as it may be a therapeutic solution, as you stated. However, the manuscript needs a clearer material and methods section, a clear state of the experimental unit and how you reach that number, and the addition of some experimental groups would make your conclusions stronger.
Attached you can find my comments/ concerns. Thanks.

Author Response
Dear reviewer,
Thank you for your comments and suggestions. We found them valuable and we will take them into consideration in the future studies. However, we were not able to perform any additional experiments regarding the comments, we went through them and applied in the manuscript. We hope that we answered the questions in an expected way and that all unclarities were logically explained. The answers can be found below.
Kind regards,
Authors
- Galanin endogenous and exogenous administration protected the cells from the degenerative effects of the incubation. the degenerative effects were equal to these groups and to the WT treated with copper sulfate, thus I don t think you can say that this galanin increase protected the cells against degeneration (acridine orange assay). You can say that the presence of galanin protects the cells, in opposition to a knockout.
The sentence has been changed in the line 16.
- Line 20: Don t you mean knockout instead of knockdown?
This was a mistake, which has been corrected in the line 20.
- Line 28: Again, galanin analogue NAX 27 5055, galanin overexpression and wild-type with a normal level of galanin led to lowered cell damage compared with a knockout, so the presence of galanin is the important issue for neuromast cells damage.
The sentence has been changed in the line 28.
- Line 63: delete the word many , as zebrafish has a whole genome duplication. And in the discussion, you can discuss what this difference may influence the translation of your results to humans.
The word “many” has been deleted [line 65].
- Line 68: delete how.
The word “how” has been deleted [line 73].
- In the Material and Methods section, make clear the experimental groups used and what was studied in each one, because this information is only perceived in the results/figures.
The experimental groups have been described in each section in the Material and Methods.
- Were the fish housed in a recirculating water system? Please add the pH and conductivity of the water.
These parameters have been added to the “Fish maintenance” part [line 87].
- How long the imaging took in each animal (average time)? To know for how long the animals were subjected to tricaine and if it may influence the results.
The time has been added to the “Fish maintenance” part [line 98].
- You refer that as you worked with early larvae stages, you did not require Ethic Committee permissions. However, you need permission to the creation of a transgenic line. Please acknowledge this.
The time has been added to the “Fish maintenance” part [lines 104-110]: “All fish lines are housed in the fish facility of the Laboratory of Genomics and Transcriptomics, University of Warmia and Mazury in Olsztyn, Olsztyn, Poland, which was built according with the local animal welfare standards. Studies performed on early life-stage zebrafish larvae and euthanasia do not require Ethic Committee permissions. According to the European Directive 2010/63/EU and Polish law regulations O.J. of 2015, item 266, The facility has an approval to keep and breed genetically modified zebrafish lines issued by Polish Ministry of Environment, decision number 127/2017.”
- You need to describe the number of experimental units used for each analysis, which will depend on how did you perform the incubation, i.e., how many containers did you have for group? And why that number (sample calculation)? I know that in the results you have some of that information in the text or figures, however, it is scattered, and it should be stated in this section.
This information has been added to the Materials and Methods section under “Fish maintenance” [lines 91-94]. We used a number of embryos that is sufficient for all experiments per group and give reliable results. We took into consideration that some of the eggs would not hatch or die during development.
- Line 92: add before collected.
The sentence has been corrected [line 128].
- Describe the NAX 5055 treatment before the Acridine Orange staining in the Material and Methods.
NAX 5055 treatment has been placed before CuSO4 incubation and Acridine Orange Staining [lines 113-121].
- In the NAX treatment, where were these injections made (part of the animal)? Which were the minimum and maximum volume injected? Did you use these values to determine the volume of PBS injected in the control group?
The place of injection and the volumes have been added to the Materials and Methods part in the NAX 5055 treatment [lines 116-119].
- Line 102: add with copper sulfate after incubation period.
The sentence has been changed [line 120].
- Line108: add from acridine orange after fluorescence.
The sentence has been changed [line 141].
- Please refer briefly the method you used to quantify the cell damage. Looking to the figure from the results, you can describe that you measured the number of pixels with fluorescence, thus apoptotic area, correct? In this case, did you eliminate the area from the yolk, as it has autofluorescence? Looking to Suppl. Fig 1, did you only study some trunk and terminal neuroblasts or the figure was only representative? This must be clarified in the methodology.
Explanation and description of the method have been added to the Materials and Methods section under “Quantification of fluorescent cells” section [lines 145-160].
- 17. In the RNA isolation and quantitative PCR explanation, please add also the galanin gene.
Primers sequence has been added in the table 1.
- Which method did you use to calculate the gene expression? Any correction?
The method has been added to the Materials and Methods under “RNA isolation and quantitative PCR” section [lines 168-173].
- Table 1 and line 121: Instead of ppial, didn't you mean ppia? And why did you choose this housekeeping gene, as it is not so used, and some articles describe others as more stable for zebrafish analysis?
For normalization we chose peptidylprolyl isomerase A-like (ppial), which showed no changes over the copper sulfate assay and it is widely used in zebrafish immune related research.
- In the statistical analysis, you did not used both tests to the acridine orange staining analysis (T-test was for the galanin gene expression). Please correct this. Did you first check if the data were normally distributed and if the groups data have homogeneity of variances before choosing these parametric tests? Also describe the statistic for the gene expression analysis, that I think is the same, but it is not mentioned there.
The “Statistical analysis” section has been corrected and missing information and explanation have been added [lines 179-185].
- Line 133-134: The control group was kept in PBS during the same incubation period at the same temperature. This information must be in the material and methods.
The sentence has been moved to the Materials and Methods [lines 126-127].
- Figure 1: alter the title to: Galanin expression before (CTRL) and after copper sulfate (CuSO4) treatment.
The title has been corrected in the figure 1.
- Similarly to what you did in the PCR analysis, it would be important to also have a baseline of cell degeneration using Wild Type without exposure to the copper sulfate to prove that an increase in cellular death is observed due to this agent.
These photos were taken to establish fluorescent background for the fluorescence quantification. This explanation has been added to the section “Quantification of fluorescent cells” [lines 149-151].
- Legend figure 2: Add wild-type galn+/+ before treated with galanin analogue.
The legend has been corrected in the figure 1.
- To study if galanin protects the animal against copper sulfate/ wounding, would be interesting to treat the KO with NAX to see if the degeneration profile is recovered to control levels. Is it possible to perform that and include it? Discuss this.
This would be indeed a good experiment to add to our research to show that NAX5055 can rescue the phenotype of galn-/- mutant. Unfortunately, due to the short time to address to the reviewers’ comments and changes in our scientific team we are not able to perform experiments at the moment.
- Line 150: you refer the alterations in the galanin expression. For this be important to provide the expression of galanin in the other groups (Tg, and NAX5055, as you showed the WT)? I know that it is expected that both groups have an increase in galanin expression, but it would be important to see that data and to see the difference of expression between the two groups, which could be related to the differences you showed in the expression of immune response genes.
We could have done this comparison to show the differences in galanin levels. Due to the fact that it is currently impossible for us to do additional experiments in this short time given, we added references and indications in the Introduction and Materials and Methods that these measurements have been done before and that it is confirmed that both NAX 5055 and Tg(hsp70l:galn) increase galanin levels in the zebrafish [lines: 77-79, 96].
- Line 153: according to d'Alençon et al., [25]. - this information should be in the Material and Methods section.
The reference has been moved to material and methods [line 126].
- Line 157: You refer an increase in the number of neuromasts, but didn t you count the fluorescent pixels? If the pixels were counted, the area was quantified and not the number of cells, as each cell have more than one pixel and the number of pixels can be dependent of the cell or neuromast size. Is this correct? Please clarify accordingly how we measured cell degeneration in the methodology.
Explanation of the quantification of fluorescent pixel has been added to the materials and methods under “Quantification of fluorescent cells” [lines 146 – 160].
- How did you identify the neuromasts as acridine orange is not specific for this type of cells? Did you use other specific staining such as 4-Di-2-ASP?
We used firstly copper sulfate, which is known to selectively damages the sensory hair cell population within 20 minutes and then we used copper sulfate staining method that is specific for apoptotic forms of cell death and does not significantly label cells undergoing necrotic death provoked by injury. The explanation and references have been added to the manuscript [lines 69-72].
- Line 160: after exogenous galanin please add compared with the knockout.
This sentence has been corrected [lines 221-222].
- Line 166-168: Please put the sentence We analyzed also the basal levels of these genes in the four research groups. A pattern of similar expression was visible, however without significant differences between the groups (Suppl.fig.2). after the first sentence of this section (3.3) as it is about the baseline/ groups without treatment.
The sentence has been added [lines 226-228].
- Line 168: add of copper sulfate exposure after After 40 minutes.
This sentence has been corrected [lines 232-233].
- Figure 3: clarify the groups, the figure must be self-explanatory. Clarification needed especially in the first group, galn+/+ ctrl, that I think is the WT without being exposed to copper sulfate but it is not clear.
The groups have been clarified in the legend of figure 3.
- Line 205 (end): using instead of use.
The sentence has been corrected [line 274].
- Line 222-223: The mechanism human macrophages [29]. revise this sentence as it seems incomplete.
The sentence has been corrected [line 294].
- Discuss if these results can be applied for wounding conditions in general or if the mechanism of action only works with wounding caused by copper sulfate.
This point has been added to the “Discussion and conclusion” section. [lines: 274-276]
- Suppl.fig1: Number of fluorescent pixels Describe what is the fluorescent marker and, as these are photos, you don't have the number of pixels. Please modify the legend.
The legend of suppl.fig.1 has been modified and corrected.

Reviewer 2 Report
The paper shows that deleting galanin in zebrafish larvae results in a more vigorous expression of inflammatory markers after copper lesioning of the lateral line neuromast system. Copper-mediated Degeneration of the neuromasts is inhibited by a modified galanin analog, NAX 5500, or by genetic overexpression of galanin. It is an interesting study, but I felt it would be more impactful in the context of neuromast degeneration (a model of hair cell or hearing loss), rather than the broader context of "chemical wounding".
Comments:
If the expression of the inflammatory markers is causative of neuromast degeneration it should occur before copper-induced neuromast degeneration occurs. A time course of marker expression, or at least one earlier time point just before degeneration occurs, would be worth seeing.
Could the authors explain what NAX 5055 is (a modified N-terminal fragment of Galanin with palmitoylated KKKK C-terminal extension). I am very confused about the method of injection. The methods say that the animals are injected twice- at 1hpf and 28 hpf. It then mentions an incubation period in which the larvae were kept in “egg water” that “contained 20uM of NAX 5500”. How long is the incubation- is this the time between the two injections or until the CuSO4 is added at 3dpf? How does the injection work- is the injection intracellular and if so would the injected NAX reach the receptor-binding site, which is facing outwards from the plasma membrane (i.e. there is membrane barrier between the NAX inside cells and the receptor binding site outside the cells). In the 28hpf injection, what part of the larvae is injected? It is very unclear. Since the larvae are incubated in the NAX solution is there any need for NAX injections?
The Neuromast degeneration assay is not clearly explained- These are lateral line cells that show specific susceptibility to copper toxicity. The assay is not, therefore as the paper says “chemical wounding”, which implies a generalized injury, but instead a specific chemical lesioning of the lateral line sensory system.
The supplementary material mentions a video, but there is no video link in the supplementary and no reference to the video in the text.
There are many minor errors in English, especially in the discussion.
Author Response
Dear reviewer,
Thank you for your comments and suggestions. We found them valuable and we will take them into consideration in the future studies. However, we were not able to perform any additional experiments regarding the comments, we went through them and applied in the manuscript. We hope that we sufficiently answered to Your comments and corrected all typos in the manuscript. Below You can find our answers to the comments.
Kind regards,
Authors
- If the expression of the inflammatory markers is causative of neuromast degeneration it should occur before copper-induced neuromast degeneration occurs. A time course of marker expression, or at least one earlier time point just before degeneration occurs, would be worth seeing.
It would be indeed worth seeing, unfortunately we did not perform any measurements at any earlier time point. Due to the short time to address to the reviewers’ comments and changes in our scientific team we are not able to perform experiments at the moment, but we will definitely take it into consideration in the future.
- Could the authors explain what NAX 5055 is (a modified N-terminal fragment of Galanin with palmitoylated KKKK C-terminal extension). I am very confused about the method of injection. The methods say that the animals are injected twice- at 1hpf and 28 hpf. It then mentions an incubation period in which the larvae were kept in “egg water” that “contained 20uM of NAX 5500”. How long is the incubation- is this the time between the two injections or until the CuSO4 is added at 3dpf? How does the injection work- is the injection intracellular and if so would the injected NAX reach the receptor-binding site, which is facing outwards from the plasma membrane (i.e. there is membrane barrier between the NAX inside cells and the receptor binding site outside the cells). In the 28hpf injection, what part of the larvae is injected? It is very unclear. Since the larvae are incubated in the NAX solution is there any need for NAX injections?
NAX5055 explanation has been added in the “Introduction” section [lines: 75-79]. The injections have been explained in the “NAX 5055 treatment” in Materials and Methods section [lines 116-119].
- The Neuromast degeneration assay is not clearly explained- These are lateral line cells that show specific susceptibility to copper toxicity. The assay is not, therefore as the paper says “chemical wounding”, which implies a generalized injury, but instead a specific chemical lesioning of the lateral line sensory system.
The explanation of the assay has been corrected in the Materials and Methods section and in the introduction. We also changed term “chemical wounding” in the title and the manuscript, to make it more clear and associated with lateral line damage and chemically induced inflammation.
- The supplementary material mentions a video, but there is no video link in the supplementary and no reference to the video in the text.
There is no video in the supplementary materials, this information has been deleted.
- There are many minor errors in English, especially in the discussion.
The manuscript has been reviewed and errors have been corrected.
Round 2
Reviewer 1 Report
Dear authors,
Thanks for your effort answering all my questions. Apart from minor comments, I still ask you to clarify the experimental unit and the rationale for using that number (also in comments below), as this is an essential part of an experimental design.
Line 28: add “normal levels of” after “the presence of”.
Line 86: “pH 7.0 ±” – did you miss the value of the pH range?
Lone 112: 30 zebrafish per group? If only 30 embryos were treated with naxolone, how later you have 50 larvae to expose to CuSO4 (line 122)?
Line 117, 118: were the control groups injected with PBS twice in the yolk and then in the blood circulation, like the groups treated with NAX 5055? If not, please discuss as it may not be a full control group.
Line 179: the bars and not the error bars.
Line 221: please add “, i. e. without CuSO4 treatment” after “in the four research groups”
Line 221: delete “however”
Line 307: add after “for several conditions,” “such as chemical wounding or induced-inflammation,”
Figure 3 legend with upper case.
Regarding the letter of response:
- Concerning the number of animals, you wrote that “We used a number of embryos that is sufficient for all experiments per group and give reliable results.” How did you know this number was enough? That it is why is important to present the rationale of sample calculation. What do you mean with 3 independent experiments? 3replicas, meaning that you did, for example, three containers of control animals exposed to CuSO4 (and the same per each group?). if so, that is the experimental unit and not the number of larvae.
- Which subunit of the peptidylprolyl isomerase A-like did you identify? Please add it.
- In question 29, you stated “…and then we used copper sulfate staining method that is specific for apoptotic forms”. Did you mean acridine orange?
Author Response
Dear reviewer,
Thank you for your comments and suggestions. The changes have been corrected in the text. We also tried to answer the questions in a clear way. You can find our comments and answers in this document below.
Kind regards,
Authors
- Line 28: add “normal levels of” after “the presence of”.
This sentence has been corrected [line 28].
- Line 86: “pH 7.0 ±” – did you miss the value of the pH range?
The value has been added [line 86].
- Line 112: 30 zebrafish per group? If only 30 embryos were treated with naxolone, how later you have 50 larvae to expose to CuSO4 (line 122)?
This is a mistake, the number has been corrected [line 122]. It was 50 embryos per group.
- Line 117, 118: were the control groups injected with PBS twice in the yolk and then in the blood circulation, like the groups treated with NAX 5055? If not, please discuss as it may not be a full control group.
The information about the control group has been corrected [line 117 – 118]. The control group has been treated in the same way as the studied group.
- Line 179: the bars and not the error bars.
This sentence has been corrected [line 180]
- Line 221: please add “, i. e. without CuSO4 treatment” after “in the four research groups”
This sentence has been corrected [line 221].
- Line 221: delete “however”
This sentence has been corrected [line 222].
- Line 307: add after “for several conditions,” “such as chemical wounding or induced-inflammation,”
This sentence has been corrected [line 307].
- Figure 3 legend with upper case.
The legend in the figure 3 has been corrected.
- Regarding the letter of response:
Concerning the number of animals, you wrote that “We used a number of embryos that is sufficient for all experiments per group and give reliable results.” How did you know this number was enough? That it is why is important to present the rationale of sample calculation. What do you mean with 3 independent experiments? 3replicas, meaning that you did, for example, three containers of control animals exposed to CuSO4 (and the same per each group?). if so, that is the experimental unit and not the number of larvae.
For one experiment we used 50 embryos per group. We repeated the experiment three times with the same amount of embryos. In total approximately 150 embryos per group was used. This number was based on calculations made with the help of power analysis (G*Power 3.1.9.7 Software). One for t-test and another for one-way ANOVA. This information has been added to the Statistic section in the Material and Methods [lines 179 – 180].
Which subunit of the peptidylprolyl isomerase A-like did you identify? Please add it.
It is peptidylprolyl isomerase Ab (cyclophilin A), if I understand the question correctly. This information has been added to the RNA isolation and quantitative PCR section in the Materials and Methods [line 171 – 172].
In question 29, you stated “…and then we used copper sulfate staining method that is specific for apoptotic forms”. Did you mean acridine orange?
That’s correct. Copper sulfate incubation selectively damages neuromasts and acridine orange is used to selectively stain apoptotic forms of cell death without necrotic forms.
Reviewer 2 Report
I thank the authors for addressing most of my original questions and for successfully correcting the small errors in English in the first version.
In the original comments I asked about “ the context of neuromast degeneration.” It is still not clear that loss of neuromasts following CuSO4 incubation is an inflammatory process (as opposed to specific toxicity of neuromasts that triggers a secondary inflammatory reaction). It would help contextualize this;what does the literature say about inflammation and the CuSO4 assay? Reference 26 (d’Alencon et al., BMC Biol. 2010; 8: 151) reported that “ Exposure of fish larvae to sublethal concentrations of copper sulfate selectively damages the sensory hair cell population inducing infiltration of leukocytes to neuromasts within 20 minutes.” This implies that damage to the hair cells occurs first (and is presumably prevented by galanin) and the cell damage then activates the inflammatory reaction. If this is, in fact, the case it should be stated to avoid possible misunderstanding.
Author Response
Dear reviewer,
Thank you for your comment. In the literature the assay with copper sulfate is described as a trigger to oxidative stress, which is followed by cell death and inflammation. It means that you are right with the implication that the damage to the hair cells occurs first, as a toxic effect of copper sulfate, and then the inflammatory reaction is activated. We clarified this information in lines 72 – 76. We hope that it this clarifies the neuromast degeneration and connected with it secondary inflammatory response.
Kind regards,
Authors
In the original comments I asked about “ the context of neuromast degeneration.” It is still not clear that loss of neuromasts following CuSO4 incubation is an inflammatory process (as opposed to specific toxicity of neuromasts that triggers a secondary inflammatory reaction). It would help contextualize this;what does the literature say about inflammation and the CuSO4 assay? Reference 26 (d’Alencon et al., BMC Biol. 2010; 8: 151) reported that “ Exposure of fish larvae to sublethal concentrations of copper sulfate selectively damages the sensory hair cell population inducing infiltration of leukocytes to neuromasts within 20 minutes.” This implies that damage to the hair cells occurs first (and is presumably prevented by galanin) and the cell damage then activates the inflammatory reaction. If this is, in fact, the case it should be stated to avoid possible misunderstanding.